# Modeling and Prediction of Thermophysiological Comfort Properties of a Single Layer Fabric System Using Single Sector Sweating Torso

**DOI:** 10.3390/ma15165786

**Published:** 2022-08-22

**Authors:** Farzan Gholamreza, Yang Su, Ruoyao Li, Anupama Vijaya Nadaraja, Robert Gathercole, Ri Li, Patricia I. Dolez, Kevin Golovin, René M. Rossi, Simon Annaheim, Abbas S. Milani

**Affiliations:** 1School of Engineering, University of British Columbia, Kelowna, BC V1V 1V7, Canada; 2Lululemon Athletica, Vancouver, BC V6J 1C7, Canada; 3Department of Human Ecology, University of Alberta, Edmonton, AB T6G 2N1, Canada; 4Department of Mechanical & Industrial Engineering, University of Toronto, Toronto, ON M5S 3G8, Canada; 5Empa, Swiss Federal Laboratories for Materials Science and Technology, Laboratory for Biomimetic Membranes and Textiles, 9014 St. Gallen, Switzerland

**Keywords:** thermophysiological comfort, computational model, single layer fabric, COMSOL Multiphysics

## Abstract

Thermophysiological comfort is known to play a primary role in maintaining thermal balance, which corresponds to a person’s satisfaction with their immediate thermal environment. Among the existing test methods, sweating torsos are one of the best tools to provide a combined measurement of heat and moisture transfer using non-isothermal conditions. This study presents a preliminary numerical model of a single sector sweating torso to predict the thermophysiological comfort properties of fabric systems. The model has been developed using COMSOL Multiphysics, based on the ISO 18640-1 standard test method and a single layer fabric system used in sportswear. A good agreement was observed between the experimental and numeral results over different exposure phases simulated by the torso test (R^2^ = 0.72 to 0.99). The model enables a systematic investigation of the effect of fabric properties (thickness, porosity, thermal resistance, and evaporative resistance), environmental conditions (relative humidity, air and radiant temperature, and wind speed), and physiological parameters (sweating rate) to gain an enhanced understanding of the thermophysiological comfort properties of the fabric system.

## 1. Introduction

Over the past few decades, significant advances have been made in the sportswear industry to develop athletic apparel that possess numerous characteristics that enhance personal comfort. Among all the comfort factors, thermophysiological comfort plays a primary role in maintaining thermal balance, which corresponds to personal satisfaction with a person’s immediate thermal environment [1,2,3,4,5]. A physically active person generates heat that needs to be dissipated to the environment to maintain thermal balance. Perspiration (sweating) also protects against overheating by dissipating heat from the skin through evaporation. The ability of dry and wet heat to dissipate from the skin depends on the properties of the trapped air between the skin and the garment (air gap), the clothing textile layers, and the boundary air layer at the outer surface of the garment. Failure to dissipate heat and moisture from the body to the environment may result in physiological strains such as heat stress and heat exhaustion for athletes, which can affect their health and performance [6,7].

In the early days, human wear trials were the only available testing method to evaluate the thermophysiological comfort of textiles due to the lack of objective test methods [5]. The development of testing devices such as sweating hot plates, sweating cylinders, and thermal manikins has enabled the prediction of the thermophysiological comfort of textiles using bench-scale and full-scale test methods [8,9,10,11,12]. Laboratory test methods are cost and time effective compared to human wear trials. Among the test methods developed, manikins and sweating cylinders combined with Fiala’s physiological model offer high accuracy for evaluating the physiological impact of a fabric system in a series of simulated activities [13,14,15]. This has led to the development of the standard test methods ISO 18640-1 and ISO 18640-2, enabling the prediction of coupled heat and moisture transfer and physiological responses without the need to prototype and manufacture clothing before testing [16].

The requirements for ethical approval and limitations, high cost, time-intensiveness of human trials, as well as the limitations of the laboratory test methods have urged the need for numerical simulations to predict the thermophysiological comfort properties of fabric systems [17]. Mathematical models have been developed and are widely used to gain a better understanding of the comfort properties of clothing. Henry created one of the earlier models that simulated the coupled effects of moisture and heat diffusion in textiles [18]. Henry’s model showed that simplifications such as modeling the fabric structure as a porous medium with defined physical properties such as porosity and permeability can be made and still produce an accurate result. Models have been improved by considering the combination of air and fibers in the porous structure of the fabric systems, the homogenous microclimate (air gap) between the fabric and the skin, and the contribution of radiative heat transfer between the fibers [19,20]. In later studies, mathematical models were further developed by incorporating the effect of moisture transport [21,22,23,24,25,26]. The development of numerical simulation has advanced our understanding of the critical properties of fabric systems, the environment, and the clothing microclimate that affect moisture and heat transfer. Fabric properties such as thickness, porosity, thermal conductivity, emissivity, and surface diffusivity, as well as ambient temperature and wind speed largely influence the thermal and evaporative resistance of a fabric system, which represent the thermophysiological comfort of fabrics [25,26]. Theoretical studies have also looked at the effect of the microclimate on heat and moisture transfer through fabric systems. An increase in the air gap thickness was found to increase the thermal and evaporative resistance of fabric systems [26]. However, the increase in the air gap thickness above a certain value may result in natural convection and lead to a decrease in the thermal insulation of the fabric system [23,25]. More advanced numerical models investigated heat transfer in the air gap between the clothing and the skin. These models examined dynamic air gaps with oscillating clothing layers to incorporate the pumping effect caused by body motion and external forces, [27] and the effect of spatial heterogeneity of enclosed air layers [17,28,29,30].

### Research Gap and Objective of the Present Work

Recently, multiphysics simulation tools such as FLUENT, ABAQUS, and COMSOL Multiphysics have gained popularity. Such tools are now often used to model fabric systems and predict the desired physical properties or functionality of textiles [31,32,33]. COMSOL Multiphysics allows the simulation of multiple coupled physical properties such as the coupling of heat and moisture transport in fabrics. This tool provides an accessible Graphic User Interface (GUI) and multiple physics interfaces that enable the modeling and simulation of different physical processes using finite element analysis. Recent studies have used COMSOL Multiphysics to create numerical models to simulate the isothermal diffusion of water vapor through textiles and predict fabric’s comfort properties [32,33,34,35]. These studies reported fabric physical properties such as air permeability, thermal resistance (R_ct_), evaporative resistance (R_et_), and water vapor diffusion as indicators of textile comfort [36,37]. However, these models did not take into account the coupled effects of heat and moisture transport. The physical properties of a wet fabric are different from that of a dry fabric and are largely influenced by moisture evaporation and condensation in the fabric [38,39]. In addition, a flat configuration of fabric is typically not representative of clothing being worn, as a chimney effect may develop in the vertical cylindrical system corresponding to most human body activities, which then leads to increased heat loss [40]. Therefore, the numerical simulation of sweating torsos would be desirable to measure the combined heat and moisture transfer in fabrics using non-isothermal conditions.

In this study, we develop a preliminary COMSOL Multiphysics numerical model of a single sector sweating torso with a single layer fabric system. The numerical model represents coupled heat and moisture transfer and simulates liquid transport in the porous fabric structure. The simulation results are compared with measurements conducted according to the ISO 18640-1. The model is then used to investigate the effect of fabric properties (fabric thickness, porosity, evaporative resistance), environmental conditions (relative humidity and wind speed), and physiological parameters (sweat rate) on the thermophysiological comfort properties of the fabric system.

## 2. Material and Methods

### 2.1. Fabrics

Fabrics employed for this study are commercially available fabrics used in sportswear (Lululemon Athletica, Vancouver, BC, Canada). The fiber content and the structural features of the selected fabrics are listed in Table 1.

### 2.2. Fabric Physical Property Measurement

The physical properties of the fabrics were measured under standard test conditions (20 ± 2 °C, 65 ± 5% RH). The results are shown in Table 2. The mass of each fabric was measured according to ASTM D3776/D3776M-20 under 1 kPa [41]. The fabric thickness and the fabric air permeability were measured according to ASTM D1777-96(2019) and ASTM D737-18, respectively [42,43]. The thermal resistance (R_ct_) and evaporative resistance (R_et_) of each fabric system were determined according to ISO 11092 [44].

The porosity ($\varepsilon_{p}$) of fabrics refers to the void fraction or the total void spaces within fibers, yarns, and fabrics [45,46,47]. It can be estimated according to Equation (1):(1)$$\varepsilon_{p}=1-\phi =1-\frac{\rho_{f}}{\rho_{\mathrm{fiber}}}$$
Here, the packing factor (ϕ) is the ratio of the fabric density ($\rho_{f}$) to the fiber density, $\rho_{\mathrm{fiber}}$. A packing factor closer to 1 indicates a fabric with less air in its structure and, therefore, low porosity. The fabrics studied here consist of blended yarns. Therefore, the density of fibers in the fabric was estimated as the weighted density based on the fabric’s fiber content (Table 1) [48]. The density of the fabric was determined by dividing the mass of the fabric by its thickness [49]. The values of the estimated porosity of the fabrics are shown in Table 2.

### 2.3. Measurement with the Single Sector Sweating Torso

The fabric systems were tested with a Single Sector Sweating Torso (Empa, St. Gallen, Switzerland) according to ISO 18640-1 [16]. The torso is an upright cylindrical apparatus simulating the human trunk. The torso consists of three layers of polytetrafluoroethylene (PTFE), high density polyethylene (HDPE), and aluminum to simulate the human skin layers with similar thermal properties (Figure 1). The sweating cylinder is also equipped with 54 independently-controlled sweating nozzles distributed on the torso to provide 0.01 sweating nozzles per cm^2^. A wicking layer with good symmetrical wicking properties (MMT wetting time for top and bottom <3.5 s, MMT maximum wetted radius for top and bottom > 20 mm) and negligible thermal insulation (R_ct_ < 0.015) is used to evenly and symmetrically spread moisture on the torso. This layer closely simulates perspiration on the human skin (50 to 250 sweat glands per cm^2^) [16]. This wicking layer is not laminated with a semi-permeable membrane and wetting of the fabric is expected to investigate the measurement of the combined heat and moisture transfer. The torso is operated inside a climatic chamber. The cylinder is heated electrically using three heating foils. The temperature of the PTFE layer is measured by a nickel wire. The heated guards at the top and bottom of the torso avoid axial heat loss. The torso is mounted on a scale to determine the evaporated and condensed amount of water. More details on the sweating torso can be found in ISO 18640-1 [16].

The experiments were performed by the Swiss Federal Laboratories for Material Science and Technology (Empa). The fabrics were washed once according to EN ISO 6330 at 40 °C [50] and were conditioned for at least 8 h at 20 ± 0.2 °C and a relative humidity of 50 ± 5%. For the measurement, each fabric was tightly wrapped around the torso without creases, leaving no air gap between the torso surface and the fabric. The climatic chamber was set to provide a temperature of 20 ± 0.2 °C, a relative humidity of 50 ± 5%, and a 1 ± 0.25 m/s wind speed. Each test was performed in three phases according to the ISO 18640-1 standard [16]: Acclimation, Simulated activity, and a Rest phase. The conditions for each of these phases are described in Table 3. During phase 1, the surface of the torso was set at a constant temperature (35 °C) with no sweating for 60 min, to determine the thermal resistance of the dry fabric. Phase 2 was conducted at a constant heating power of 125 W (288 W/m^2^) and constant sweating, which released 100 g of water over a 60 min period at a constant flow rate. This simulates the physical activity of a human at a medium intensity of 6 MET (Metabolic Equivalents of Task). This phase provides characteristic parameters about the cooling performance of the fabric during physical activity including cooling delay (CD; time until a negative trend of Torso surface temperature is observed), initial cooling (IC, rate of temperature reduction observed after onset of cooling), sustained cooling (SC, rate of temperature reduction during quasi-steady temperature course) and moisture update (accumulated moisture in the fabric at the end of phase 2). Phase 3 represents the recovering phase of a human after physical activity (1 MET), and involves a constant heat rate of 25 W (58 W/m^2^) with no sweating for 60 min. The drying time of the fabric is determined gravimetrically during this phase (time to stabilization of Torso weight). The temperature and the weight courses were recorded during these three phases of the test.

## 3. Numerical Modeling

### 3.1. Geometric Model

The geometric model included three parts: the chamber, the fabric, and the torso. They were simulated according to the measurements provided in ISO 18640-1 [16]. It was assumed that the entire system is symmetrical with respect to the *x*-axis (Figure 2). As such, a simple 3D symmetrical model of the torso, fabric, and chamber was considered to reduce the computation time. The chamber and the surrounding air were modeled by a rectangular block. The chamber dimensions were chosen to provide a wind speed of 1 m/s with laminar flow and constant environmental conditions (temperature and relative humidity). Only a portion of the ambient air needed to be modeled, since the heat and moisture transfer mainly occurs close to the fabric. Additionally, the airflow above and beneath the torso was neglected in the model as the wind flow was in the positive x-direction. The fabric was sandwiched between the torso and the chamber and was modeled on a macroscopic scale. The wicking layer covering the sweating torso, described in the experimental approach section, was not modeled. However, it is assumed that the water is evenly distributed over the entire torso to simulate the human skin sweating and the resulting combined heat and mass transfer process. The thickness of the fabric was adjusted according to the values in Table 1 for Fabrics A to D. The dimensions of the geometric model of the torso and the chamber are shown in Figure 2.

### 3.2. Material Properties

The single layer fabrics were assumed to be a homogeneous medium and modeled as a porous material. The materials defined in COMSOL Multiphysics for the torso and the climatic chamber corresponded to the materials specified in ISO 18640-1 [16]. The thicknesses of the fabrics used are provided in Table 2. The thermal conductivity, $k_{f}$ [W/(m·K)], of the fabrics was determined according to Equation (2) [44], where $t_{f}$ is the thickness of the fabric [m] and R_ct_ $\left[{\;m}^{2}\cdot K/W\right]$ is the thermal resistance of the fabric determined according to ISO 11092 [44].
(2)$$k_{f}=\frac{t_{f}}{R_{\mathrm{ct}}}$$

The diffusion coefficient of water vapor through the fabric, $D_{eff}\;\left[m^{2}/s\right]$, was calculated using Equation (3), where $\tilde{R}$ [J/(kg·K)] is the universal gas constant of water vapor, $M_{w}$ [g/mol] is the molecular weight of water, and $\lambda_{w}\;\left[\mathrm{kJ}/\mathrm{kg}\right]$ is the heat of vaporization of water at temperature T [K] [51]. The evaporative resistance, $R_{\mathrm{et}}\;\left[\mathrm{Pa}\cdot m^{2}/W\right]$ was measured according to ISO 11092 [44].
(3)$$D_{eff}=\frac{t_{f}\tilde{R}T}{R_{\mathrm{et}}{\;\lambda }_{w}M_{w}}$$

### 3.3. Physics Setup 

This study neglected the radiation heat transfer and the thermal contact resistance between the torso surface and the textile. It was also assumed that the temperature and humidity of the fabrics were uniform in the initial conditions. In this model, the sweating nozzles were simplified as a boundary with a specific simulated sweat rate. In addition, the three consecutive phases were modeled according to the experimental approach and the requirements to assess the thermophysiological properties of fabrics described in Table 3.

The airflow in the entire numerical domain was solved for steady state. The results were then used to compute the transient transport of heat and mass. The steady-state momentum equation for the air in the ambient domain is
(4)$$\rho \left(u\cdot \nabla \right)u=\nabla \cdot \left[-pI+\mu \left(\nabla u+{\left(\nabla u\right)}^{T}\right)\right]$$
where u is the velocity vector [m/s], ρ is the fluid density $\left[\mathrm{kg}/m^{3}\right]$, p is the pressure of the fluid [Pa], μ is the dynamic viscosity of the fluid [kg/(m·s)] and I is the identity matrix. No body force is included in the model. Based on the assumption of incompressible flow, the mass conservation is governed by
(5)∇·u=0

In the porous domain, the airflow is modeled using the Brinkman Equation [32]:(6)$$\frac{\rho }{\varepsilon_{p}\;}\left(u\cdot \nabla \right)\frac{u}{\varepsilon_{p}\;}=-\nabla p+\nabla \cdot \left[\frac{\mu }{\varepsilon_{p}}\left(\nabla u+{\left(\nabla u\right)}^{T}\right)\right]-\frac{\mu }{K}u$$
where K is the air permeability. For the airflow in the fabric, the mass conservation is satisfied by Equation (5). The boundary condition at the inlet of the domain is u·n=1 m/s, and the pressure at the outlet remains at 1 atm. No-slip boundary conditions are applied to the solid surfaces.

The transport of vapor in the fabric and ambient environment is governed by [52]
(7)$$\frac{\partial c_{v}}{\partial t}+\nabla \cdot J_{v}+u\cdot \nabla c_{v}=\frac{Q_{m}}{M_{w}}$$
Here, $c_{v}$ $\left[\mathrm{mol}/m^{3}\right]\;$ is the mole concentration of vapor, u is the velocity of air from Equations (4)–(6), and $Q_{m}$ is a source term related to evaporation. In the fabric, the air velocity u is from Equation (6), and the vapor diffusive flux $J_{v}$ is
(8)$$J_{v}=-D_{eff}\nabla c_{v}$$

For vapor transport in the ambient, $Q_{m}=0$, u is from Equation (4), and $J_{v}$ is given
(9)$$J_{v}=-D\nabla c_{v}$$
where D is the diffusion coefficient of vapor in air.

The vapor concentration of the free stream ambient airflow can be determined based on free stream conditions (relative humidity RH = 50% and ambient temperature $T_{\infty }=20\;℃$) by using [35]
(10)$$c_{v,\infty }=\mathrm{RH}\frac{P_{\mathrm{sat}}\left(T_{\infty }\right)}{{\tilde{R}T}_{\infty }}$$

For the saturation pressure ${\;P}_{sat}$, Tenten’s formula [53] can be used, which is
(11)$$P_{sat}=610.7\times 10^{\frac{7.5\left(T-273.15\right)}{T-35.85}}$$

Liquid water was assumed to be only present in the fabric. The transport of liquid water in the fabric is governed by
(12)$$\frac{\partial c_{w}}{\partial t}+\nabla \cdot \left(u_{w}c_{w}\right)=-\frac{Q_{m}}{M_{w}}$$
Here, water concentration is evaluated using mole concentration, $c_{w}$. Saturation is important for the process of liquid water transfer. The correlation for $c_{w}$ and water saturation $S_{w}$ is defined as
(13)$$S_{w}=\frac{c_{w}M_{w}}{\rho_{w}\varepsilon_{p}}$$
where $\rho_{w}$ is the density of water. Water saturation $S_{w}$ is the ratio of the water-occupied pore volume to the total pore volume. In Equations (12) and (14), the capillary diffusion due to the gradient of $S_{w}$ is not considered.

The velocity of water was calculated using Darcy’s Law [54], which is
(14)$$u_{w}=-\frac{KK_{wr}}{\mu_{w\;}}\nabla p$$
Here, $\mu_{w}$ is the dynamic viscosity of water in [Pa.s], and ∇p is the pressure gradient solved from Equation (6). $K_{wr}$ is the relative permeability of water.

The heat transfer in the porous medium physics interface was used to model the heat transfer in the fabric [32], which is governed by
(15)$${\left({\rho C}_{p}\right)}_{eff}\frac{\partial T}{\partial t}+\left({\rho C}_{p}u+\rho_{w}C_{p,w}u_{w}\right)\cdot \nabla T+\nabla \cdot q^{\prime \prime }=\dot{q}$$

In the fabric, the heat capacities and thermal conductivities of the fabric, liquid water, and moist air need to be taken into consideration [19]. In Equation (15), $C_{p}$ and $C_{p,w}$ are the specific heats of air and water, respectively. The effective heat capacity is
(16)$${\left({\rho C}_{p}\right)}_{\mathrm{eff}}=\left(1-\varepsilon_{p}\right){\;\rho }_{f}{\;C}_{p,f}+\varepsilon_{p}S_{w}\rho_{w}{\;C}_{p,w}+\varepsilon_{p}{\rho \;C}_{p}\left(1-S_{w}\right)$$
where ${\;C}_{p,f}$ is the specific heat of the fabric. The heat flux term in Equation (15) is
(17)$$q^{\prime \prime }=-k_{eff}\nabla T$$
where the effective thermal conductivity is defined as
(18)$$k_{eff}=\left(1-\varepsilon_{p}\right)k_{f}+\varepsilon_{p}{\;S}_{w}k_{w}+{\;\varepsilon }_{p}\left(1-S_{w}\right)k$$
where k and $k_{w}$ are the thermal conductivities of moist air and liquid water, respectively.

For the heat generation term $\dot{q}$ in Equation (15), evaporation needs to be determined. Evaporation occurs only when the saturation vapor concentration, $c_{v,\;sat}$, also known as the equilibrium concentration, is higher than the water vapor concentration within the fabric, $c_{v}$. Evaporation is calculated using
(19)$$Q_{m}=\gamma \left(c_{v,sat}-c_{v}\right){\;M}_{w}A_{\mathrm{spec}}$$
where γ is the evaporation rate in [m/s], and $A_{\mathrm{spec}}$ is the specific area in [$m^{2}/m^{3}$]. The saturation vapor concentration can be obtained by letting RH=1 in Equation (10), which gives
(20)$$c_{v,sat}=\frac{p_{\mathrm{sat}}\left(T\right)}{\tilde{R}T}$$

The heat generation term in Equation (15) can then be obtained from
(21)$$\dot{q}=-Q_{m}\lambda_{w}$$

### 3.4. Study Setup

Two different steps simulated the three phases of the single sector sweating torso experiment in the above COMSOL Multiphysics model. The first step was a stationary (steady-state) study that simulated and computed the airflow velocity field of the ambient air and within the fabric. This was assumed to be a constant vector field independent of the three phases. The results of this step were then used as input parameters for a time-dependent step that simulated the heat and moisture transfer in the fabric and with the surrounding air during the three phases of the experiment. These two steps were simulated separately to decrease the computation time.

## 4. Results and Discussion

A numerical model of a single sector sweating torso was developed in this study to predict the temperature and the moisture courses for the fabrics in Table 1. The experimental and numerical measurements of temperature and weight courses of the studied fabric systems are illustrated in Figure 3. The predicted values for the thermophysiological comfort properties and their definitions are given in Table 4 and Table 5.

### 4.1. General Observations

The comparison of the temperature and the moisture course trends obtained from the experimental and the numerical approaches (Figure 3) revealed that the model can predict the fabric behavior relatively well during all the three test phases (R^2^ ranged from 0.72 to 0.99). The fabrics were exposed to dry heat and a constant temperature of 35 °C for 60 min during the acclimatization phase (Phase 1) in the numerical model (Figure 3). In phase 2 (sweating zone), initial cooling and sustained cooling were observed in both experimental and numerical measurements. The trend of the cooling curve was different in Fabric C from Fabrics A, B, and D. This could be attributed to the cellulosic fibers in Fabric C having relatively higher absorption regains in high moisture concentration and delayed onset of cooling than the synthetic fibers used in the other fabrics (A, B and D) (Figure 3a,b) [55]. The initial cooling occurred at a slower rate in Fabric C compared to the other fabrics, while the sustained cooling was more pronounced. Fabric C consists of 95% cellulosic fibers, which may cause very slow diffusion in the dry state (lower IC values compared to synthetic fibers), but it becomes much more rapid at high regains. Further, the increased moisture content lowered the thermal resistance of the fabric (higher SC values compared to synthetic fibers) [56,57]. This also explains the cooling delay observed in Fabric C during the initial sweating phase. In addition, a slight increase in the temperature was observed at the onset of sweating (cooling delay) in Fabric D, made of synthetic fibers with nylon as major fiber content in its structure (82%). Diffusion is very slow in nylon in the dry state compared to the polyester and elastomeric fibers in Fabrics A and B [57]. This also explains why no cooling delay was observed in Fabrics A and D. The heat of sorption, i.e., the heat evolved when water is absorbed by the textile at a given moisture regain, is absorbed by the human body and, consequently, increases the skin temperature [58,59], leading to cooling delays (Figure 4c). Furthermore, the heat of wetting is significantly higher in cellulosic fibers than the synthetic fibers used in the studied fabrics. Nylon has a relatively higher heat of water absorption among the synthetic fibers. [58,60]. The differences between the cooling delays in Fabrics C and D therefore clarifies the effect of the heat of sorption in the wetting of textiles (Figure 3a,b).

Post cooling (PC) was observed in the simulation results in the four studied fabrics, and a similar trend was observed in the experimental measurements. However, the model had lower measurements of initial cooling (IC) and sustained cooling (SC) and higher measurements for post cooling (PC) for Fabric D compared to the experimental values. Fabric D’s cooling behavior was significantly affected by its initial condition (cooling delay and initial cooling) in the numerical simulation (Figure 4d). The model underestimated the cooling delay and initial cooling, which led to lower values of the temperature course during Phase 2. This could be due to the sensitivity of the model to porosity and thickness (mentioned in Section 4.3.1). Fabric D had the lowest porosity and highest thickness. That can affect the initial conditions (Phase 2 from time = 60 to approx. 75 min) and the sustained cooling (Phase 2 from time = approx. 90 to 120 min). However, the difference observed for the sustained cooling value, when compared to the experimental value, was in the same range as observed for the other fabrics. 

By comparing the moisture course curves of the four fabrics in the experimental (Figure 3c) and the numerical (Figure 3d) measurements, it was evident that the model accurately captured the fabrics’ behavior and correctly followed the trends of the experiments during the three phases. At the onset of sweating (t = 60 min), the fabrics absorbed moisture at a level based on their moisture absorption properties. Fabric C had the highest moisture uptake at the end of phase 2 (t = 120 min), when the sweating was terminated. The moisture uptake was highest in Fabric C, which can be attributed to its fiber type (95% cellulosic), fabric structure and properties (thick, heavy, and porous fabric structure), and the finish with poor wicking properties on the fabric. Fabrics A and B appeared to store less water in their structure and have the highest value of evaporated moisture after the termination of sweating (Table 4), which was also predicted by the numerical model (Table 5). Fabrics A and B showed better moisture management properties among the studied fabrics, with quick-absorbing and fast-drying fabrics observed in both experimental and numerical studies. The low retention of water in synthetic fibers such as nylon and polyester in the structure of Fabrics A and B, their thin and light structure, and the wicking finishes could be the reason behind their greater moisture management properties [59,61].

### 4.2. Model Predictability 

The statistical similarity between the numerical model and experimental results (R^2^_CD_ = 0.98, R^2^_IC_ = 0.72, R^2^_SC_ = 0.99, R^2^_MU_ = 0.98, and R^2^_DT_ = 0.98) clearly indicates the capability of developed model to predict the thermophysiological comfort properties of the fabrics. The model can predict thermophysiological comfort properties with 0 to 34% difference (except the initial cooling of Fabric D). Figure 4 and Figure 5 show these results for individual fabrics. The thermophysiological comfort properties of Fabric C: (cooling delay (CD), initial cooling (IC), sustained cooling (SC), moisture uptake (MU), and drying time (DT)), are depicted in Figure 4c and Figure 5c as an example [16]. Superscripts “E” and “N” are used to distinguish between experimental and numerical approaches. 

### 4.3. Parametric Study

The development of the above model allowed investigating the effect of fabric properties (thickness, porosity, evaporative resistance), environmental conditions (relative humidity, ambient temperature, wind speed), and the physiological parameters (sweat rate) on the thermophysiological comfort properties of the fabric systems. In this analysis, the parametric study was run on the fabric system during phases 2 (sweating zone) and 3 (post sweating zone). 

#### 4.3.1. Effect of Fabric System Properties

The effect of fabric system properties on the moisture course measurements was investigated. For a given set of environmental conditions and physiological parameters, the effect of each fabric property was obtained for phases 2 and 3, while the other properties of the fabric remained constant. 

The effect of thickness is depicted in Figure 6a. The increase in the thickness of the fabric can cause a significant increase in moisture uptake, which results in more moisture accumulation in the fabric system (Figure 6a). At the onset of phase 3, the thickest fabric had the highest condensed moisture in its structure. As such, more energy needed to be consumed to evaporate the moisture in the fabric, which prolonged its drying time. These model predictions agree well with previous research [26,38].

Changes in the thermal resistance of the fabric had only a minor effect on the moisture course of the fabric systems (Figure 6b). At the onset of sweating (phase 2, t = 60 to t = 90 min), the fabric with higher porosity had slightly less resistance to evaporation in comparison with fabrics with lower porosities (Figure 6c). This is in agreement with previous studies where lower values of evaporative resistances were observed when the fabric porosity approached 1 [26]. It was also observed that at a specific time during sweating (phase 2, t = 90 to t = 120 min), the fabric with the higher porosity had more condensed water in its structure than fabrics with lower values of porosity. However, high porosity resulted in an increase in the moisture uptake (Figure 6d, phase 3, t = 120) in the fabric system once sweating was terminated, as well as a significant increase in the drying time compared to the fabrics with lower porosity (Figure 6d, phase 3, t = 120 to t = 180 min).

#### 4.3.2. Effect of Environmental Conditions and Sweat Rate

Figure 7a,b illustrate the effect of ambient relative humidity and temperature on the amount of water in the fabric systems, respectively. The curves of the moisture course show that activities in dryer and warmer environments can decrease the condensed water, moisture uptake, and drying time of the fabrics compared to ambient conditions with higher relative humidity and lower temperatures. More evaporation in the fabric systems was observed as the ambient wind speed was increased, as expected. The airflow transitioned from natural convection to forced convection as the wind speed was increased from 0.1 to 2 m/s, which enhances evaporation in the system and decreases moisture uptake and drying time (Figure 7c). 

In phase 2, the constant heating power of 125 W (288 W/m^2^) and constant sweating for 60 min is recommended by the ISO 18640-1 standard in order to release 100 g of water per hour continuously. This testing condition simulates physical activity at a medium intensity of 6 MET. Here, sweating rates of 50 g/h and 200 g/h were chosen to simulate a human carrying out low (4 MET: 190 W/m^2^) or high intensity (10 MET: 470 W/m^2^) exercise, respectively [62]. The model predicted that the increase in the sweating rate to 200 g/h increases the moisture uptake and the drying time of the fabric substantially (Figure 7d). The numerical model predicted zero dripped water during the sweating phase, meaning that the delivered water from the cylinder was either absorbed by the fabric or evaporated. This further confirmed that the fabric system exhibits excellent moisture management properties under the defined sweating rates (50 to 200 g/h). Overall, the developed model is a helpful tool to predict the physiological comfort property of the fabric system under moderate to intensive physical activities.

## 5. Conclusions

The simulation of a single sector sweating Torso was carried out using COMSOL Multiphysics, where the thermophysiological comfort properties of fabric systems were predicted. The trend of the temperature course and the moisture course obtained from the experimental and the numerical investigations showed that the model is capable of predicting the fabric behavior during the three phases (R^2^ ranged from 0.72 to 0.99). In particular, the model could predict the properties of fabrics related to thermophysiological comfort including CD, IC, SC, moisture uptake, and the drying time. Through the subsequent sensitivity analysis, the model predicted that the porosity and fabric thickness are the most dominant factors affecting the thermophysiological comfort properties of the tested fabric systems.

## Figures and Tables

**Figure 1 materials-15-05786-f001:**
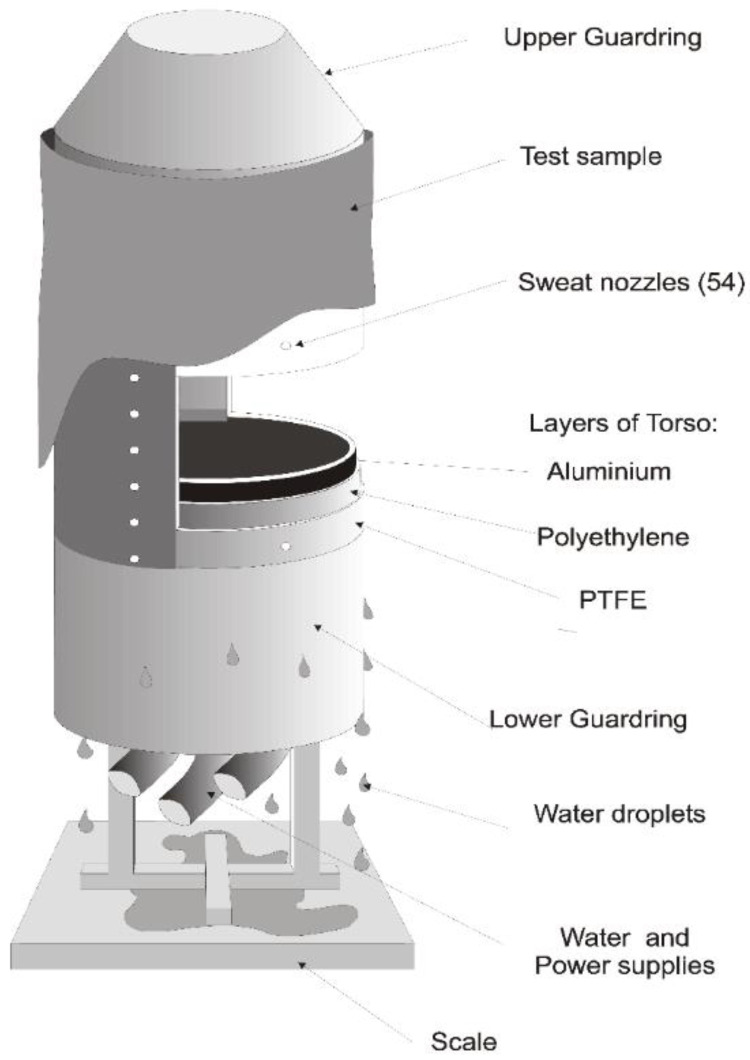
Schematic illustration of sweating torso.

**Figure 2 materials-15-05786-f002:**
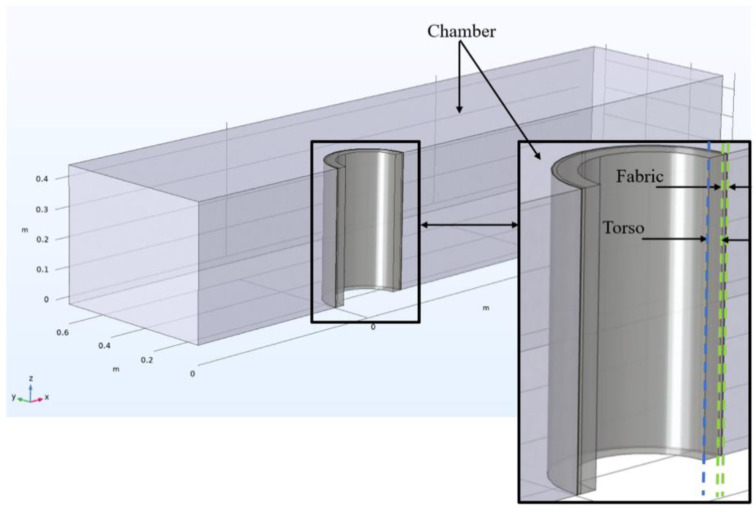
Geometry of the model representing an upright cylindrical apparatus simulating the human trunk in a climatic chamber.

**Figure 3 materials-15-05786-f003:**
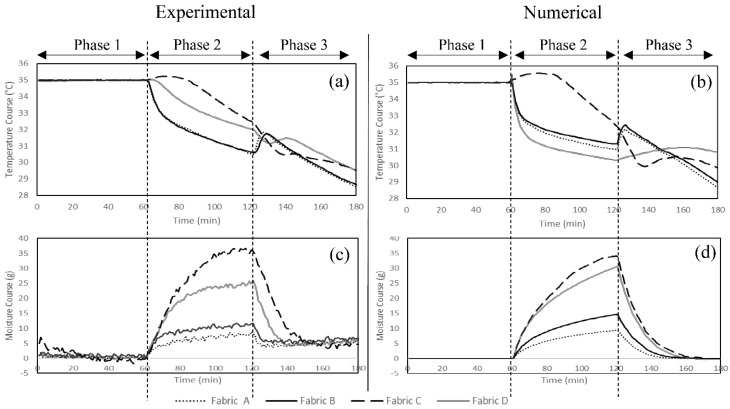
(**a**) Experimental and (**b**) numerical measurements of the temperature course and (**c**) experimental and (**d**) numerical measurements of the moisture course of the fabric systems during the 3 phases.

**Figure 4 materials-15-05786-f004:**
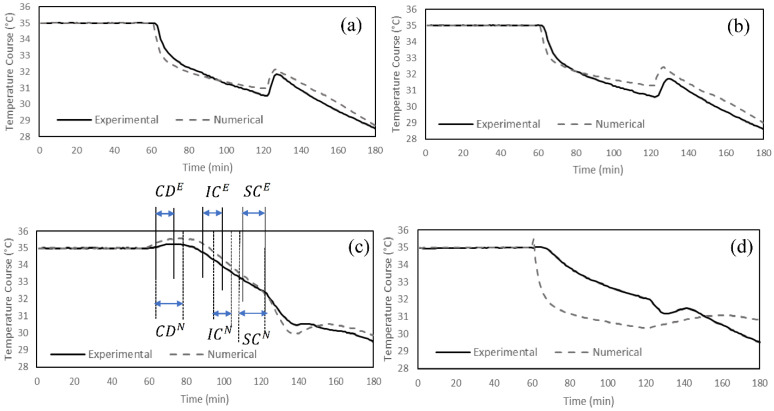
Temperature course during 3 phases of measurements for (**a**) Fabric A, (**b**) Fabric B, (**c**) Fabric C, and (**d**) Fabric D.

**Figure 5 materials-15-05786-f005:**
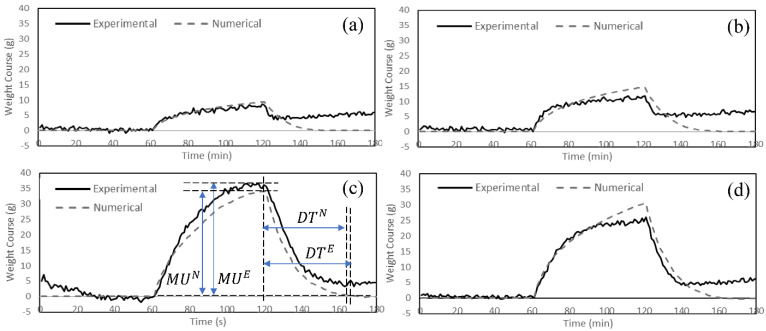
Weight course during 3 phases of measurements for (**a**) Fabric A, (**b**) Fabric B, (**c**) Fabric C, and (**d**) Fabric D.

**Figure 6 materials-15-05786-f006:**
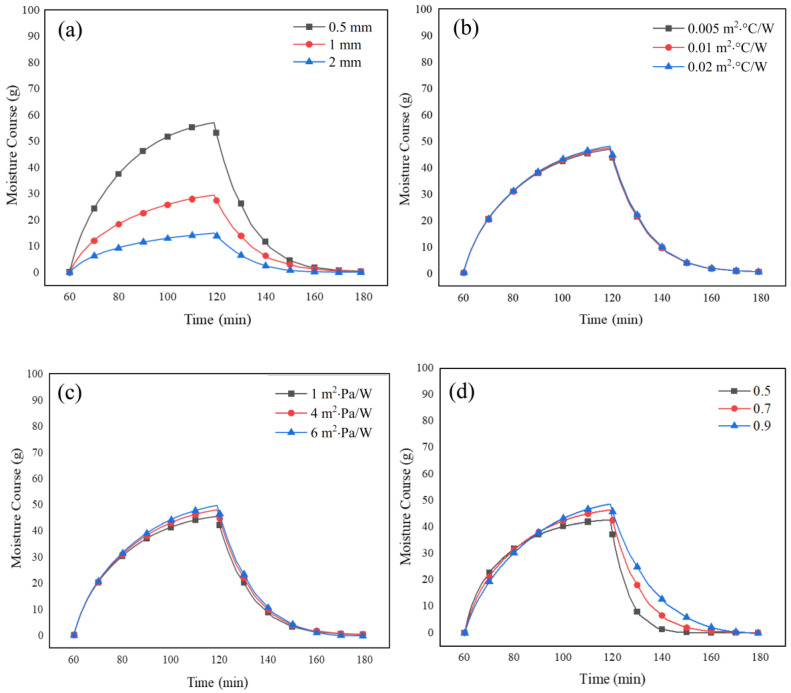
The effect of (**a**) fabric thickness, (**b**) thermal resistance (**c**) evaporative resistance, and (**d**) fabric porosity on moisture course measurement obtained from the numerical model.

**Figure 7 materials-15-05786-f007:**
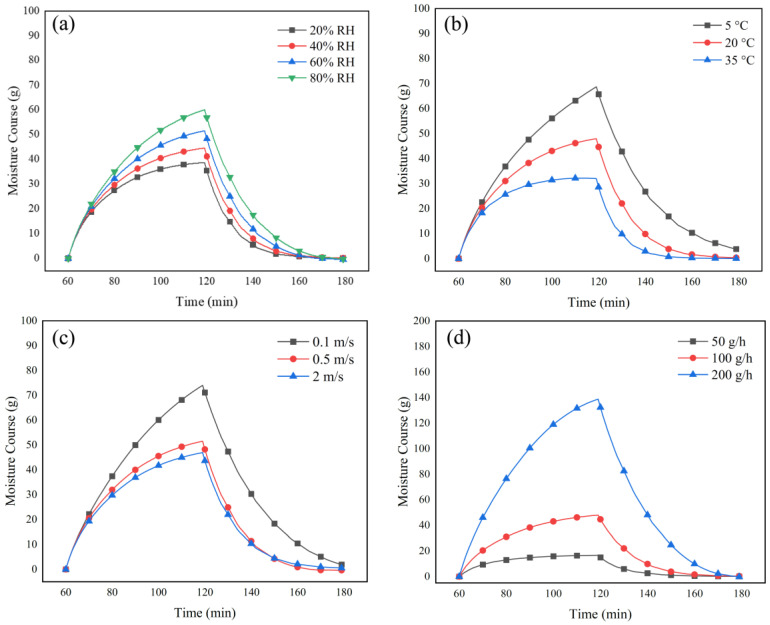
The effect of environmental conditions ((**a**) relative humidity, (**b**) ambient temperature, and (**c**) wind speed) and (**d**) sweat rate on the moisture course measurement obtained from the numerical model.

**Table 1 materials-15-05786-t001:** The fiber content and the structural features of the fabric systems.

Fabrics	Fiber Content	Fabric Structure	Surface Property
Fabric A	86% Polyester, 14% Elastane	Plain weave (woven)	Wicking finish
Fabric B	51% Nylon, 38% Polyester, 8% Elastane, 3% X-static	Single jersey (knit)	Wicking finish
Fabric C	71% Pima cotton, 24% Lyocell, 5% Elastane	Single jersey (knit)	Water repellent finish
Fabric D	83% Nylon, 17% Elastane	Single jersey (knit)	Wicking finish

**Table 2 materials-15-05786-t002:** Physical properties of the fabrics.

Fabrics	Mass (g/m^2^)	Thickness (mm)	Fabric Density (Kg/cm^3^)	Air Permeability (cm^3^/cm^2^/s)	Porosity	R_ct_ (m^2^·K/W)	R_et_ (m^2^·Pa/W)
Fabric A	127	0.30 ± 0.0	420	9.7	0.69	0.063 ± 0.0	0.78 ± 0.01
Fabric B	97	0.41 ± 0.01	240	43.8	0.86	0.074 ± 0.001	2.05 ± 0.13
Fabric C	180	0.60 ± 0.02	300	33.8	0.79	0.079 ± 0.001	4.10 ± 0.43
Fabric D	328	0.79 ± 0.02	420	8.1	0.64	0.073 ± 0.001	3.58 ± 0.17

**Table 3 materials-15-05786-t003:** Standard test profile representing the three consecutive phases and the requirements to assess thermophysiological properties of fabrics [16].

Phase Number	Phase Name	Duration (min)	Phase Condition	Sweat Rate (g/h)	Standard Evaluation
1	Acclimation	60	Constant temperature 35 °C	0	Dry thermal insulation (R_ct_)
2	Simulated activity (6 MET)	60	Constant heating power of 125 W	100	Moisture management and cooling properties
3	Rest phase (1 MET)	60	Constant heating power of 25 W	0	Post cooling and drying behavior

**Table 4 materials-15-05786-t004:** The experimental results of phase 2 of the standard experiment according to ISO 18640.

Fabrics	Cooling Delay (CD) (min)	Initial Cooling (IC) (℃/h)	Sustained Cooling (SC) (℃/h)	Moisture Uptake (MU) (g)	Dripped Moisture (g)	Evaporated Moisture (g)	Drying Time (DT) (min)
Fabric A	0.0 ± 0.0	16.4 ± 0.8	3.0 ± 0.8	8.1 ± 2.1	0.0	89.9 ± 2.0	2.7 ± 0.6
Fabric B	0.0 ± 0.0	16.6 ± 0.6	1.7 ± 0.4	10.4 ± 1.5	0.0	87.6 ± 1.5	3.7 ± 1.2
Fabric C	12.0 ± 2.0	1.9 ± 0.4	3.8 ± 0.0	36.5 ± 0.7	0.0	61.6 ± 0.8	27.0 ± 0.9
Fabric D	3.2 ± 0.1	5.1 ± 0.6	2.0 ± 0.2	24.3 ± 0.8	0.0	73.7 ± 0.8	15.3 ± 0.6

Cooling Delay (CD) is the time until the temperature of the fabric decreases at the beginning of phase 2. Initial Cooling (IC) is the rate of change in temperature of the initial part of cooling after the CD. Sustained cooling (SC) is the rate of change in temperature during the steady-state part of phase 2 after the IC. Moisture Uptake (MU) is the amount of moisture in the fabric after phase 2. Dripped Moisture is the amount of water that runs off the torso and is collected by a hydrophobic cloth. Evaporated Moisture is the difference between the given off moisture, moisture uptake, and the dripped moisture [16].

**Table 5 materials-15-05786-t005:** The numerical results of phases 2 and 3. “Difference” refers to the differences between the simulated and experimental (Table 4) values. The negative/positive differences show when the model underestimates/overestimates the fabric properties.

Fabric Properties	Fabric A		Fabric B		Fabric C		Fabric D	
	Numerical Value	Difference (%)	Numerical Value	Difference (%)	Numerical Value	Difference (%)	Numerical Value	Difference (%)
Cooling Delay (CD) (min)	0	0	0	0	15	−25	2	33
Initial Cooling (IC) (℃/h)	19	−15	18.4	−11	2	−7	15	>100
Sustained Cooling (SC) (℃/h)	1.4	34	1.3	34	5.3	−23	1.5	33
Moisture Uptake (MU) (g)	8.6	−6	12.6	−25	33.5	8	33.2	−34
Drying Time (DT) (min)	1.9	32	2.9	21	28.0	−1	20	−32

## Data Availability

The data presented in this study are available on request from the corresponding author.

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
