# Peer review of "Modeling and Prediction of Thermophysiological Comfort Properties of a Single Layer Fabric System Using Single Sector Sweating Torso"

_materials, 2022, doi:10.3390/ma15165786_

Round 1

Reviewer 1 Report

Page 4, Table 2: where possible the  used units should the SI units (fabric density in kg/m2).

Kelvin degree is defined as K only, not as °K.

 Figure 1 – Sweating torso: the distribution of the sweating nozzless is not clear. Are these nozless during the measurement by the heated cylinder covered with a cotton fabric with the surface semi-permeable membrane, which distributes the water fed through the nozzles evenly over the surface of the cylinder and simultaneously prevents liquid water from passing to the outside while letting wated vapou pass though (from Performance of protective clothing , ASTM 1997)?

In this case the Sweating torso simulates the so-called dry sweating  - no liquid water enters directly the analyzed fabrics, accumullated water only may appear inside of the these fabrics.  

HOWEVER, THE MANUSCRIPT DOES NOT EXPLAIN CLEARLY, WHETHER THIS MEASUREMENT MODE IS USED IN THIS CASE.

If there is no separating membrane used, then water enters the studied fabrics directly and thermal and evaporation resistances of the studied fabrics will will be affected substantively and non-uniformly (see papers by Mangat, Boguslawska and Hes). Then, the applied mathematical model may not work properly.

Mangat, M. M., Hes, L., Bajzík, V. (2015). Thermal resistance models of selected fabrics in wet state and their experimental verification. Textile Research Journal, 85(2), 200-210.

Bogusławska-Bączek M, Hes L. Effective Water Vapour Permeability of Wet Wool Fabric and Blended Fabrics. FIBRES & TEXTILES in Eastern Europe 2013; 21, 1(97): 67-71.

Reviewer 2 Report

This article entitled “Modeling and prediction of thermophysiological comfort properties of a single layer fabric system using single sector sweating torso” developed a preliminary numerical model which can combine the coupled effects of heat and moisture transport o to predict the thermophysiological comfort properties of fabric systems. The whole work is fundamental and meaningful. Please double check the whole paper carefully and revise some minor issues according to the following comments.

1.      Layers of Torso consists of polytetrafluoroethylene (PTFE), high density polyethylene (HDPE) and aluminum, what’s the basis for choosing these materials? Additionally, the simplified conditions and innovation for establishing the preliminary COMSOL Multiphysics numerical model of a single sector sweating torso with a single layer fabric system need to be pointed out.

2.      The manuscript should make some further analysis of the reason why the experimental temperature course for Fabric D in Phase 2 in Figure 3(a) is obviously higher than that in numerical measurements in Figure 3(b), and for Fabric D, why there is an opposite trend of Phase 3 in experimental and numerical measurements.

3.      In Table 5, error refers to the differences between the simulated and experimental (Table 2) values. There should be corrected as Table 4, right? In addition, there is no experimental data about drying time.

4.      It’s better to add the unit of the relative humidity in Figure 7.
